# Using Landsat OLI and Random Forest to Assess Grassland Degradation with Aboveground Net Primary Production and Electrical Conductivity Data

**Hao Yu** [1,2], **Lei Wang** [3], **Zongming Wang** [2], **Chunying Ren** [2,*] **and Bai Zhang** [2]

1   School of Geomatics and Prospecting Engineering, Jilin Jianzhu University, Changchun 130118, China; yuhao@jlju.edu.cn or xiaoxiaoyupiu@163.com

2   Key Laboratory of Wetland Ecology and Environment, Northeast Institute of Geography and Agroecology, Chinese Academy of Sciences, Changchun 130102, China; zongmingwang@iga.ac.cn (Z.W.); zhangbai@neigae.ac.cn (B.Z.)

3   Department of Geography and Anthropology, Louisiana State University, Baton Rouge, LA 70803, USA; leiwang@lsu.edu

*   Correspondence: renchy@iga.ac.cn; Tel.: +86-431-8554-2297

**Abstract:** Grassland coverage, aboveground net primary production (ANPP), and species composition are used as indicators of grassland degradation. However, soil salinization deficiency, which is also a factor of grassland degradation, is rarely used in grassland degradation assessment in semiarid regions. We assessed grassland degradation by its quality, quantity, and spatial pattern over semiarid west Jilin, China. Considering soil salinization in west Jilin, electrical conductivity (EC) is used as an index with ANPP to assess grassland degradation. First, the spatial distribution of the grassland was measured with information mined from multi-temporal remote sensing images using an object-based image analysis combined with classification and decision tree methods. Second, with 166 field samples, we utilized the random forest (RF) algorithm as the variable selection and regression method for predicting EC and ANPP. Finally, we created a new grassland degradation model (GDM) based on ANPP and EC. The results showed the $R^2$ (0.91) and RMSE (0.057 mS/cm) of the EC model were generally highest and lowest when the *ntree* was 400; the ANPP model was optimal ($R^2 = 0.85$ and RMSE = 15.81 gC/m$^2$) when the *ntree* was 600. Grassland area of west Jilin was $609.67 \times 10^3$ ha in 2017, there were $373.79 \times 10^3$ ha of degraded grassland, with $210.47 \times 10^3$ ha being intensively degraded. This paper surpasses past limitations of excessive reliance on vegetation index to construct a grassland degradation model which considers the characteristics of the study area and soil salinity. The results confirm the positive influence of the ecological conservation projects sponsored by the government. The research outcome could offer supporting data for decision making to help alleviate grassland degradation and promote the rehabilitation of grassland vegetation.

**Keywords:** remote sensing; aboveground net primary productivity (ANPP); soil salinity; grassland degradation model (GDM); random forest (RF); principle component analysis (PCA)

---

## 1. Introduction

Grasslands make entire ecosystems more resilient to environmental changes by preventing soil erosion and regularizing water regimes, particularly in arid/semiarid regions [1,2]. Area and spatial distribution information of grasslands is important for understanding plant, animal, and bird species survival [3]. Grasslands also play a vital role in sustaining human lives [4]. Grassland degradation is the process of retrograde succession and productivity decline of the grassland ecosystem under unreasonable utilization (overgrazing and irrational reclaiming) and unusual natural processes

(drought, wind sand, water erosion, saline-alkali, waterlogging etc.) [5,6]. The main manifestations are the height, coverage, yield and quality of grassland vegetation, the deterioration of soil habitat, and the decline of production capacity and ecological function [7]. In the past several decades, the degradation of grassland caused not only the decline of productivity of grassland itself, but also the deterioration of ecological environment and the threat to human survival and development. In recent decades, grassland degradation has become a severe global environmental problem (e.g., endangering regional ecosystem services and functions) [8–10]. At present, about 90% of grasslands in China undergo degradation and 34% of rangelands are moderately to severely degraded [11,12]. Human activities can completely convert grasslands to other land functions [13], which could turn fertile soil into barren lands (e.g., saline-alkali land, sand) that can no longer yield products [14–16]. Measuring the degradation of grassland ecosystems is an urgent issue for both the government and local communities, especially for semiarid and arid regions of China where ecosystems are more fragile.

Grassland degradation is conventionally studied through field investigation, but this method is time-consuming and costly in large-scale research. Remote sensing offers numerous technologies to monitor grassland degradation [17–19]. Most research on grassland degradation has been carried out by measuring changes of grassland area [20,21], aboveground vegetation condition [22,23], and other characteristics of grasslands [24]. The assessment methods of grassland degradation can be divided into two categories: Visual classification and degraded index inversion. The former classifies the levels of grassland degradation according to the characteristics of imagery, which has a higher demand for classification experts in the professional aspect and is time-consuming and costly compared with the latter. With the emergence of massive remote sensing data, the latter is more widely used for grassland degradation assessment. Zhang et al. used Normalized Differential Vegetation Index (NDVI) data (8 km spatial resolution) to evaluate grassland degradation over the Mongolia Plateau [25]. Li et al. estimated grassland aboveground biomass using multi-temporal MODIS data (250 m spatial resolution) in the West Songnen Plain, China [22]. Tarantino et al. detected the changes in semi-natural grasslands by cross correlation analysis with WorldView-2 images and Landsat 8 data [26]. Wang et al. monitored grassland degradation through effective spectral feature parametrization methods exactly distinguished the constructive species and degraded indicators using hyperspectral remote sensing data [27]. Liu et al. assessed the grassland degradation from Landsat TM image in conjunction with grass cover and proportion of unpalatable grasses (PUG) [24]. In review of previous researches, although the low spatial resolution sensors have high temporal resolution, which enables a good overview of grassland dynamics, they are not suitable for obtaining the details of area change and monitoring degraded levels at regional scale [28]. High spatial resolution imagery has affluent detail information and prominent texture information of terrestrial object which compensates for the spatial limitation, but it could not be widely used for spatial assessment of grassland degradation at large scale due to its small coverage and expensive cost [29]. Landsat 8 Operational Land Imager (OLI) has a spatial resolution of 30 m. In addition, there is a panchromatic band with a resolution of 15 m, which could fulfill the demands for monitoring grassland degradation. Therefore, the medium spatial resolution was extensively applied for the evaluation of grassland degradation [23,30,31].

In the selection of degradation indicators, previous studies evaluated the degraded level by vegetation cover, vegetation biomass, aboveground net primary production (ANPP), PUG and grassland species. Aboveground net primary production is the energy value that producers can use for growth, development, and reproduction, and it is also the material basis for the survival and reproduction of other biological members in the ecosystem. Moreover, ANPP is a parameter that can reflect environment changes because it is sensitive to biological and abiotic factors [32]. Variability in ANPP has been used as an indicator to measure the loss of ecosystem function due to grassland degradation in several studies [33,34]. These studies took full account of the grassland situation above the ground, but did not consider the soil circumstance of grassland. Moreover, grassland salinization, a common phenomenon due to the buildup of salts in soil, is the primary form of grassland degradation in arid, semi-arid, and dry sub-humid areas [35,36]. West Jilin Province of China is a typical semiarid

region that lies between saline-alkali land and wind-blown sand land. The low-lying terrain and poor drainage blocked the waterways from releasing salts, consequently causing soil salinization and grassland degradation [37,38]. The research shows that the area of soil salinization accounts for 34% of West Jilin Province. About 77% of grassland areas were salt-affected [39]. The grasslands in this area used to be good pasture with high yield and good grass quality, but due to uncontrolled exploitation and utilization of land resources, the vegetation cover has been seriously degraded in recent years. Previous research established that soil salinization was a factor in grassland degradation in semiarid and arid regions [37,40]. This has been accompanied by serious salinization; the grassland degradation has also been enhancing itself through a negative feedback mechanism caused by land cover change and soil salinization [37,41]. Yu et al. (2018) reported that grassland was the most severely affected by salinity among all vegetation land covers in West Jilin Province. Other authors studied salinization characteristics of saline and alkali grassland soils in semiarid regions and analyzed the differences in grassland patterns with different salinity and alkali levels [41]. The growth of vegetation is directly affected by the soil properties. It is necessary to establish an evaluation system of grassland degradation based on grassland biomass and soil salinity in semiarid regions and then analyze the spatial distribution of degraded grassland combined with auxiliary data in the study area [42,43]. In previous studies, more attention has been paid to aboveground biomass, plant coverage, and species of vegetation than soil properties in estimating grassland degradation. Therefore, this paper aims to establish a new grassland degradation model (GDM), designed for semiarid regions and based on the spatial distribution of ANPP and soil salinity to grassland degradation measured from satellite imagery.

Random forest (RF) has been widely used in vegetation biomass and soil parameter estimation in recent years [44,45]. The RF is a popular ensemble learning method that can be used to establish predictive models for both classification and regression problems using the random uncorrelated decision tree [46]. Compared with general regression analysis, RF has very fast speed of processing, and it performs well in dealing with big data. The RF does not need to worry about the multivariate collinearity problem. RF method is not sensitive to noise in training samples. Therefore, the accuracy of models could be much higher compared with other machine learning and the traditional statistical regressions [47]. Here, the inversion models of ANPP and electrical conductivity (EC) were established by the RF obtaining the relationship between remote sensing imagery' spectral information and field samplings.

This research has the following objectives: (1) measure the grassland extent and their spatial distribution over west Jilin Province in 2017; (2) formulate the relationships between reflectance bands, spectral indices, and field measurements of ANPP and EC using RF; and (3) and establish the grassland degradation model by principle component analysis (PCA) using the predicted ANPP and EC. We expect that the outcome of this research could offer a remote-sensing-based grassland degradation monitoring methodology for long-term land resource management in semiarid regions.

## 2. Materials and Methods

### 2.1. Materials

#### 2.1.1. Study Area Descriptions

The west Jilin Province (43°59′~46°18′ N, 121°38′~126°11′ E), one of the most serious saline-alkali areas in the world [48], is located in the southern Songnen Plains in northeast China. The elevation of the study area ranges from 100–200 m above sea level [49]. The area is influenced by temperate, semi-arid continental monsoon climate. The mean annual precipitation is about 411 mm, which increases gradually from west to east. The mean annual evaporation is about three times as much as precipitation and increases from east to west. High evaporation and low precipitation are major factors for soil salinization over west Jilin Province [39,50]. Average winter temperature is −16 °C and summer is 23 °C. Winds are generally moderate year round, averaging 3–6 m/s [51,52].

Grassland is the most widely distributed land cover type except cropland in this area. The species of grass are *Leymus chinensis*, *Phragmites australis*, *Leymus secalinus*, *Spodiopogon sibiricus*, *Cleistogenes polyphylla*, *Chloris virgate*, and *Iris lactea var. chinensis*. During past decades, the proportion of *Leymus chinensis* decreased, but the proportion of unpalatable grassland increased with the increasing prominence of ecological environment problems. The soils are Chernozem (Haplic Chernozem, FAO, Rome, Italy), meadow soil (EutricVertisol, FAO, Rome, Italy), aeolian soil (Arenosol, FAO, Rome, Italy), Solonetz (Solonetz, FAO, Rome, Italy), and Chestnut soil (Haplic Kastanozem, FAO, Rome, Italy). The widespread fragmented, scar-like saline lands are part of the grassland. In the past decades, about 20% of the grasslands in the west Jilin Province have been reclaimed as farmland due to population growth and increased food demand; additionally, some regions of grasslands were depleted by overgrazing [53]. These land uses all led to the rapid reduction in the area of natural steppes [54]. The phenomenon of grassland degradation is noticeable, seriously restricting the development of both natural ecology and social economy. Figure 1 shows the location of the study area.

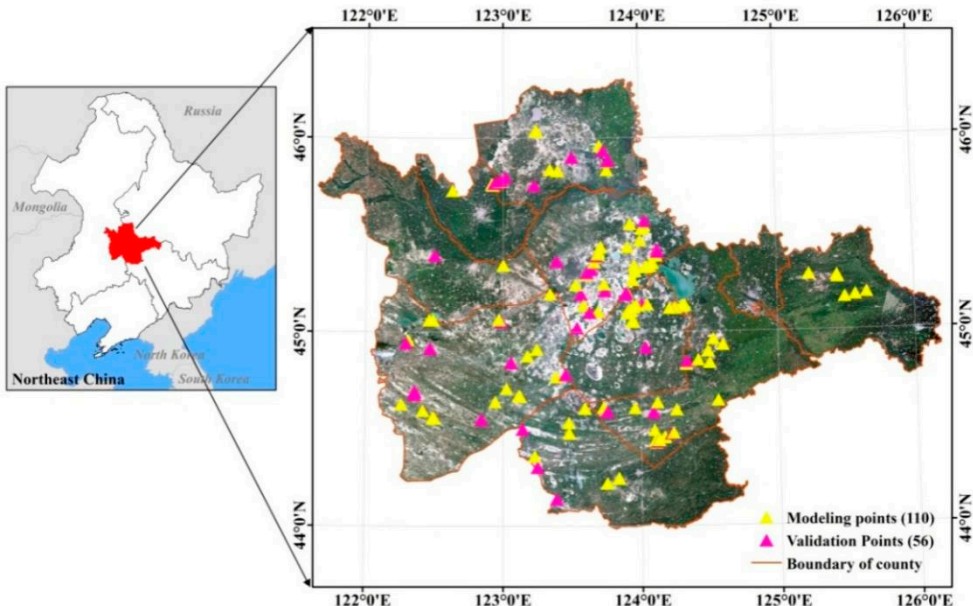

**Figure 1.** Locations of study areas and the samplings' locations mentioned in this paper, with true color composite image of Landsat Operational Land Imager (OLI) in 2016 and 2017.

### 2.1.2. Remote Sensing and Pre-Processing

We used a total of six scenes from Landsat 8 OLI (https://www.usgs.gov/) and two Sentinel-2A scenes (https://scihub.copernicus.eu/) from the year 2017. The sentinel-2A scenes were only used to verify the accuracy of land cover type classification in 2017. Most of the image data for this study is cloud-free and acquired during similar seasons (August–September). The spatial resolution is 30 m, which is sufficient for grassland and other land use analysis [49]. Geo-rectification, atmospheric correction, and radiometric calibration were applied to each image prior to interpretation to correct additive and multiplicative atmospheric effects; band reflection and spectral indices could be calculated after processing [55]. In order to better understand the relationship between spectral information and measured data in the field, we selected eight spectral indices, which include Normalized Differential Vegetation Index [56,57], Enhanced Vegetation Index (EVI) [58], Soil Adjusted Vegetation Index (SAVI) [59], and Salinity Index (SIs) [60,61]. All spectral indices are listed in Table 1.

**Table 1.** The expressions of spectral indices.

| Spectral Index | Expression | Full Name |
|---|---|---|
| NDVI | (NIR − Red)/(NIR + Red) | Normalized Difference Vegetation Index |
| DVI | NIR − Red | Different Vegetation Index |
| EVI | NIR/Red | Enhanced Vegetation Index |
| SAVI | (NIR − Red)/(NIR + Red + 0.5) × 1.5 | Soil Adjusted Vegetation Index |
| SI | $\sqrt{\text{Green} \times \text{Red}}$ | Salinity Index |
| SI2 | $\sqrt{\text{Green}^2 + \text{Red}^2 + \text{NIR}^2}$ | Salinity Index2 |
| SI3 | $\sqrt{\text{Green}^2 + \text{Red}^2}$ | Salinity Index3 |
| SI4 | SWIR1/NIR | Salinity Index4 |

### 2.1.3. Field Data and Samplings' Processing

Soil and grassland biomass samples were collected from 170 sampling sites located in grassland with different status. Three 0.5 m$^2$ plots at peak biomass were selected in each area (10–21 August 2016 and 8–19 August 2017); green and recently dead grass material at the soil surface was clipped and placed in paper bags [62]. After obtaining the grassland biomass, soil samples were collected from the 0–5 cm layer under the corresponding grassland. All grassland samples were dried for at least 48 h at 65 °C and then weighed. In order to more effectively compare with previous studies, grassland biomass (g) units are expressed in the form of carbon (gC) [35,63]; biomass and the conversion coefficient (0.45) were used to measure ANPP according to Piao et al. (2002) and Chen et al. (2008). The expression is as follows:

$$B_g = \frac{NPP}{S_{bn}\left(1 + S_{ug}\right)} \tag{1}$$

$B_g$ represents the hay yield of grassland per unit area (g/m$^2$); $S_{bn}$ represents the conversion coefficient, the value is 0.45 and the unit is g/gC; and $S_{ug}$ represents the ratio coefficient of biomass between the aboveground and underground parts of the grassland. In this study, we only analyze the ANPP; therefore, the value was 0.

The EC was obtained by soil laboratory tests. First, each soil sample was crushed by pestle after air-drying. Second, we prepared a 1:5 soil:water suspension by passing 10 g soil through a 2 mm sieve. The suspension was mechanically shaken at 15 rpm for 1 h to dissolve soluble salts. Finally, we tested the electrical conductivity using LEICI DDS-307A meter [16,64]. Results of the test showed that four data points' values were aberrant, so we deleted the data from these four sampling points when establishing the inversion models. Currently, 166 valid sampling points have been divided into two parts which included a training dataset (n = 116) and a validation dataset (n = 50).

Auxiliary data was used to support the interpretation and result analysis. Digital Elevation Model (DEM, resolution 30 m) data was used for classification and geo-rectified. The reference/validation sites for classification of land cover in 2017 were obtained from field survey, random extraction from Sentinel-2A, and from Google Earth.

### 2.2. Methods

#### 2.2.1. Object-Based Image Analysis (OBIA) and Decision Tree

Object-Based Image Analysis (OBIA) divides the image into non-overlapping objects; there is no gap between objects, which is in accord with the topological relationship. OBIA involves pixels being grouped into objects based on either spectral similarity of images or an external variable such as ownership, soil type, or geological altitude [65]. An image object provides richer information, including spectral information, texture, and geometric features, as well as the relationship between different objects, such as distance [66]. There are three parameters that determine the segmentation effect: Scale, shape, and compactness [67,68]. The optimal scale of segmentation is determined by multiple

experiments and comparing effects. In this paper, the scale set to 100 for Landsat OLI could satisfy our research requirements; the value of shape and compactness parameter was 0.3 and 0.6, respectively.

Decision tree (DT) is the process of using informative rule sets to realize classification [69]. The rules are easy to understand and the classification process is also in line with the cognitive process of human beings [70]. The most distinctive characteristic of a decision tree is the flexibility to use multi-source data such as vector data, different spatial resolution images, and other auxiliary data (geographical, meteorological, and economic). Moreover, the decision tree could accept auxiliary data as input data with incomplete records [71]. Therefore, interpretation of grassland was completed by OBIA and DT. It is more difficult to extract grassland information compared with other vegetation types such as farmland and forestland. Thus, the distribution of grassland was extracted in two steps: (1) excluding the easily identifiable land cover types (non-vegetation, farmland, and forestland); (2) confirming the edge of the grassland according to measured data and spectral information (Figure 2).

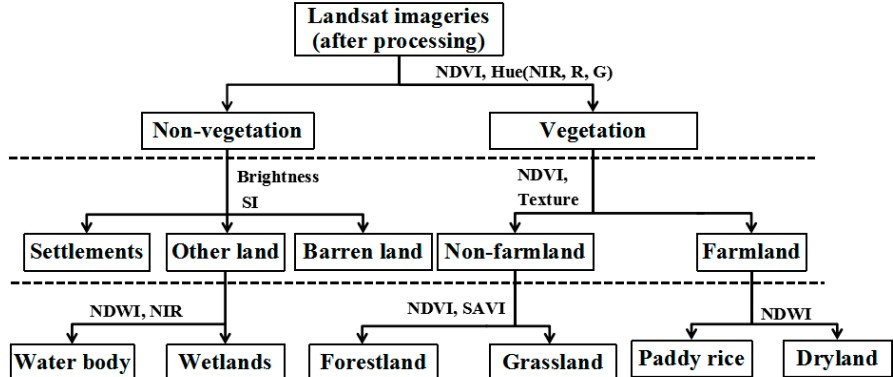

**Figure 2.** Decision-tree models for grassland classification of the west Jilin Province in 2017. (The threshold of each rule set varied with images. NIR is near-infrared band, R is red band, G is green band. NDWI is the Normalized Difference Water Index, calculated from G and NIR bands).

### 2.2.2. Random Forest Regression

RF is a powerful machine learning classifier, which has good anti-noise ability for regression performance yet has not been thoroughly evaluated in remote sensing [72,73]. RF is composed of multiple decision trees, which can be predicted by combining a series of decisions of basic models [74] improving the prediction accuracy by the average value of multiple classification decision trees on each subsample [75]. RF uses a random input subset or variables in node partition instead of the best variables, aims to reduce the generalization error, and achieves the highest accuracy [76]. Ultimately, RF algorithm has the prominent advantages of being nonparametric, prediction results with high accuracy, and the ability to measure the importance of variables compared to linear regression algorithms [73,76].

The fourteen variables used for grassland degradation assessment comprise eight spectral indexes and six spectral bands. The variables for predicting ANPP and EC were screened by RF, which excludes variables with a lower score of importance. R software package "randomForest" was used to implement inversion of ANPP and EC, and the inversion models were confirmed by three parameters: *ntree*, *mtry*, and *nodesize* [44,45]. The *nodesize* value defaulted to one, which represents the minimal size of the terminal nodes of the tree; *ntree* specifies the number of decision trees contained in a random forest; *mtry* specifies the number of variables in the node for the binary tree. Figure 3 illustrates how to establish the relationship between remote sensing imagery and field measurement to obtain the inversion models of soil salinity (EC) and ANPP inverse using RF.

The *ntree* boot samples (containing approximately one third of the elements of original dataset) were randomly extracted with playback from the original dataset; the unselected samplings comprise out-of-bag (OOB) data for testing. The prediction accuracy of each decision tree is determined by the average OOB [77]. The importance score can test the variables importance, which is calculated by the

increase percent in mean square error through permuting each variable when others are not changed. Finally, the accuracy of RF is measured by the coefficient of determination ($R^2$) and root mean square error (RMSE). In addition, model uncertainty should be carefully presented as the model output [78].

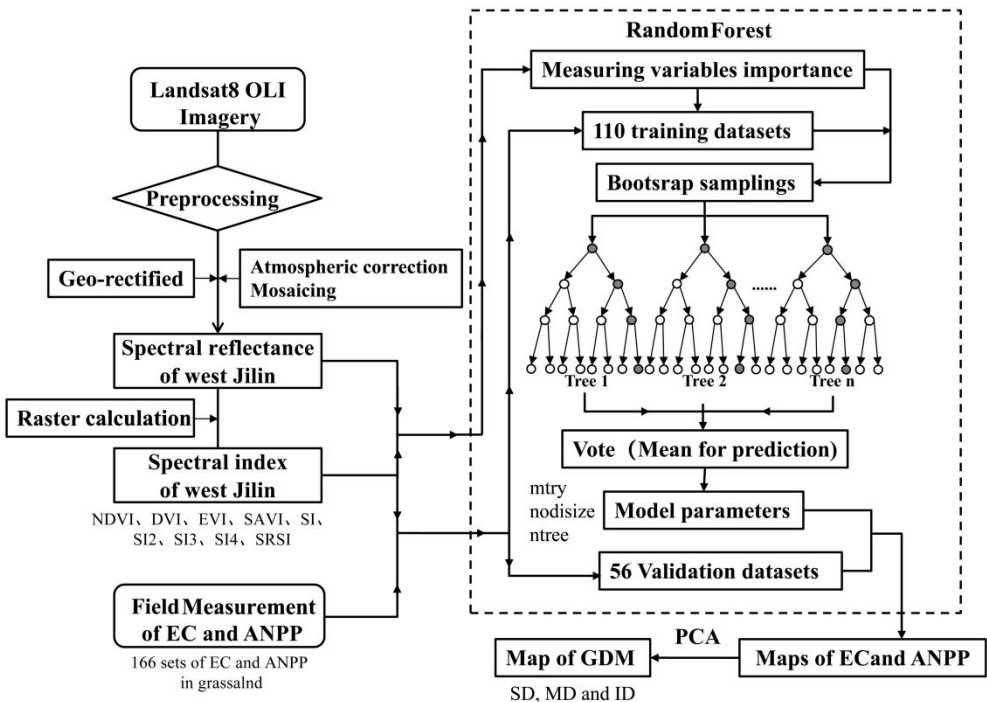

**Figure 3.** The flow-process diagram of random forest (RF) to assess grassland degradation parameters using remote sensing images and field measurements in Semiarid West Jilin, China (SD is slightly degraded, MD is moderately degraded, and ID is intensively degraded).

In our study, we not only verified the overall accuracy of the models, but also analyzed the spatial distribution of the uncertainty of the inversion models. The spatial distribution of uncertainty provides a map of the uncertainty rather than an RMSE and can therefore help understand and improve the model from a geographical aspect. The flowchart is showed in Figure 4 and the specific steps are as follows:

(1) Create a fishnet structure covering the region of grassland with a cell size of 2500 m and then extract the center point of each cell to represent the locations (Figure 4a).
(2) Assign each center point an RMSE value that is calculated based on values of sample locations. The value of the center points is the average RMSE value of multiple sampling points; the sampling points are close to the center points with a control distance (Figure 4b), which is set as 3000 m based on the area of the study area and density of sampling points in this study.
(3) Move window to the next point until all center points are processed (Figure 4c).
(4) Delete points with null value and generate the spatial uncertainty of the model by the spatial analysis tools.

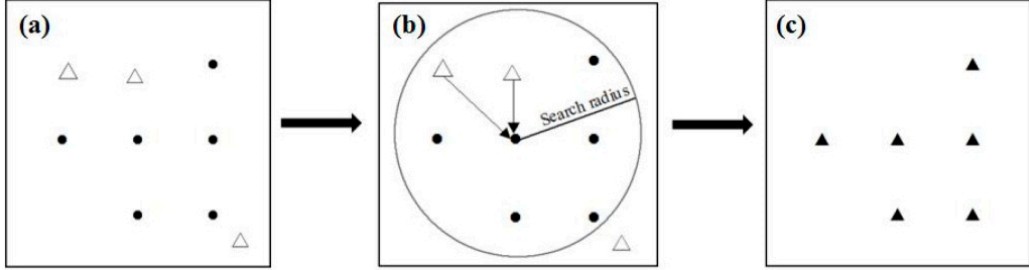

**Figure 4.** The schematic diagram of spatial uncertainty evaluation method. (**a**) extract the center points by creating a fishnet structure covering the region of grassland; (**b**) assign the sampling points' value by calculating the average value of all points within a certain distance; (**c**) complete calculation of all points in window and go to next window until finish all points.

### 2.2.3. Grassland Degradation Model from the Principal Component Analysis

Both ANPP and EC are important indicators of grassland degradation. However, the grassland degradation model must use their weighted combination, where the weights cannot be determined from empirical studies due to the lack of reference. If the weights are determined arbitrarily, it is difficult to justify or make the model generalizable. Herein, the weights of these two parameters were calculated by principal component analysis.

The principle of PCA is to attempt to recombine the original variables into a new group of independent comprehensive variables by orthogonal transformation. The transformed variables are called principal components, each one being described by an eigenvector and its eigenvalue of the covariance matrix of the input variables [79]. The principle components are combined by mutually uncorrelated variables with maximal variance. The intuitive purpose of PCA is to reduce the dimension of redundant data features, while retaining the most important part of the data features, so that the main components can maintain the integrity of all the data information. The GDM value will be the first principal component calculated by PCA method. The formula expression is as follows:

$$\mathbf{F} = \mathbf{AU} \tag{2}$$

$\mathbf{X}$ is a matric with $n$ rows and $m$ columns. $\mathbf{A}$ is an adjusted data matric, which is equal to experiment value ($\mathbf{X}$) minus the average of each column, $\mathbf{F}$ represents the new variables which are calculated by PCA, and $\mathbf{U}$ is a matric of the eigenvalue vectors which is calculated through $\mathbf{A}$'s covariance.

$$\mathbf{F} = \mathbf{AU} = \begin{bmatrix} a_{11} & a_{12} & \cdots & a_{1k} \\ a_{21} & a_{21} & \cdots & a_{2k} \\ \vdots & \vdots & & \vdots \\ a_{n1} & a_{n2} & \cdots & a_{nk} \end{bmatrix} \begin{bmatrix} \lambda_{11} & \lambda_{12} & \cdots & \lambda_{1m} \\ \lambda_{21} & \lambda_{22} & \cdots & \lambda_{2m} \\ \vdots & \vdots & & \vdots \\ \lambda_{n1} & \lambda_{n2} & \cdots & \lambda_{nm} \end{bmatrix} = \begin{bmatrix} v_{11} & v_{12} & \cdots & v_{1k} \\ v_{21} & v_{22} & \cdots & v_{2k} \\ \vdots & \vdots & & \vdots \\ v_{m1} & v_{m2} & \cdots & v_{mk} \end{bmatrix} \tag{3}$$

$\mathbf{C}$ is the matrix of parameters' coefficients which were calculated by eigenvalue and eigenvalue vector. The expression is as follows:

$$\mathbf{C}_i = \mathbf{U}_i \sqrt{Eigenvalue_i} \tag{4}$$

$i$ is the number of principle components, and $\mathbf{C}_i$ is the coefficient matrix of the parameters to the $i$ principal component of analysis result. In this part, input data were the spatial distribution of ANPP and EC, which were calculated by RF method. The first component was the output result through orthogonal transformation.

## 3. Results

### 3.1. Random Forest Modeling for EC and ANPP

RF has the ability to rank the importance of variables before predicting ANPP and EC. Using the RF method, we ranked the importance score of all variables and selected the variables with higher scores as the variables for establishing the inversion models [45]. We measured variables of importance for estimating ANPP and EC. Figure 5 shows the band Green and SI were more noticeable for the EC model, and the band Coastal, NDVI, and SAVI had more importance than other predicted variables for establishing the ANPP model. Taking this into consideration, we chose bands Green, Red, Blue, NIR and spectral indexes SI and EVI as the predicted variables for inverse EC model and bands Coastal, SWIR, Red, Blue and spectral indexes SAVI and NDVI as the predicted variables for establishing the ANPP model.

The inversion results of ANPP and EC were determined by three parameters: *nodesize*, *ntree*, and *mtry* [80]. Generally, *mtry* was tested one by one until the ideal value was found. The *ntree* can roughly judge the value of the error stability in the model by graph. Optimized parameters' value was determined by the root mean square error. The result implies that the optimal model for ANPP was established when *ntree* was 600 and *mtry* was 4; for EC, the inversion model was best when *ntree* was 400 and *mtry* was 3. $R^2$ of the ANPP and EC models was 0.85 and 0.91, and RMSE was 15.81 g/m$^2$ and 0.057 mS/cm, respectively.

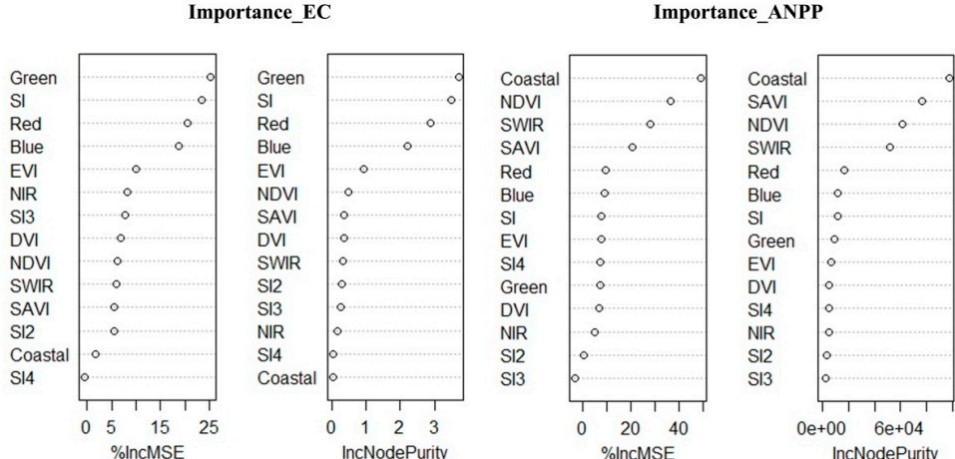

**Figure 5.** The importance score of variables for estimating aboveground net primary production (ANPP) and electrical conductivity (EC). The higher the importance scores the more important the variable for estimating ANPP and EC. SWIR is Shortwave Infrared; NIR is Near Infrared.

Additionally, we analyzed the spatial distribution of the uncertainty of the inversion models. The introduction of the method has been described in Section 2.2.2. Figure 6 shows the spatial map of uncertainty for grassland degradation parameters. Figure 6a shows that the higher RMSE of the ANPP model was distributed in the north and east of the study area. RMSE in the central region was mainly below 10 gC/m$^2$, and that in northwest of the region was above 20 gC/m$^2$. This means that the model of ANPP was more effective in the central region than the north or the east regions. Figure 6b shows that the RMSE of EC below 0.10 mS/cm nearly covered the entire area except for the central region and a small part of the north region. For the EC model, the predicted values were much higher than the measured values when the value of EC was above 1 mS/cm.

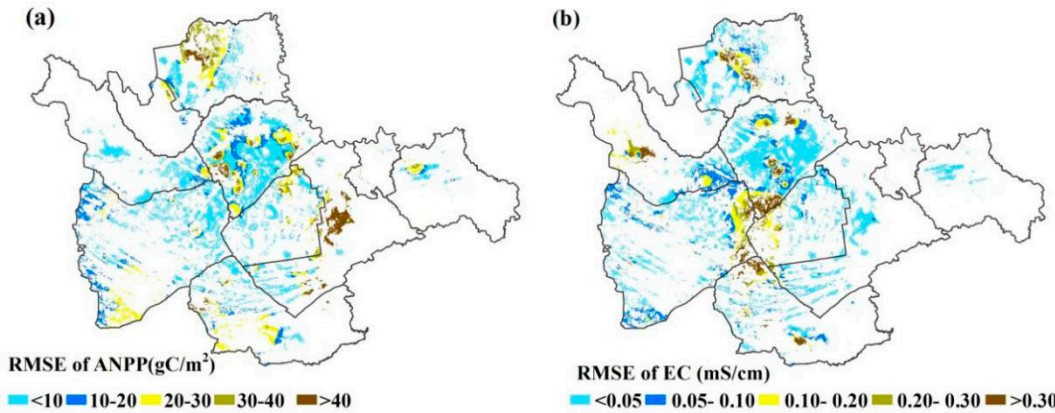

**Figure 6.** The spatial uncertainties of ANPP and EC models. (**a**) the spatial uncertainty distribution of ANPP; (**b**) the spatial uncertainty distribution of EC.

### 3.2. Spatial Distribution of ANPP and EC Using RF

The spatial distribution of grassland in west Jilin Province was obtained by OBIA and using a decision tree. The accuracy of grassland classification was assessed by field survey and random extraction points from higher resolution images. The overall grassland classification accuracy was 92.55%. Larger patches of grassland were mainly distributed in the central and north area, while smaller patches of grassland were scattered in the south of the study area. The grassland area was $606.73 \times 10^3$ ha over west Jilin Province in 2017.

The relationships between Landsat OLI imagery and sampling measurements were obtained by RF aiming to predict ANPP and EC. Figure 7 shows the spatial distribution of ANPP and EC over west Jilin Province. The Figure 7a showed that the ANPP value of grassland ranged from 32.912 gC/m$^2$ to 273.794 gC/m$^2$ and the mean value was 106.231 gC/m$^2$. 44.45% of the area had an ANPP value between 75–125 gC/m$^2$, mainly in the central and north regions. 29.79% of the area had 125–200 gC/m$^2$, concentrated in the central and eastern regions. 17.43% of the area was below 75 gC/m$^2$ located in the middle regions.

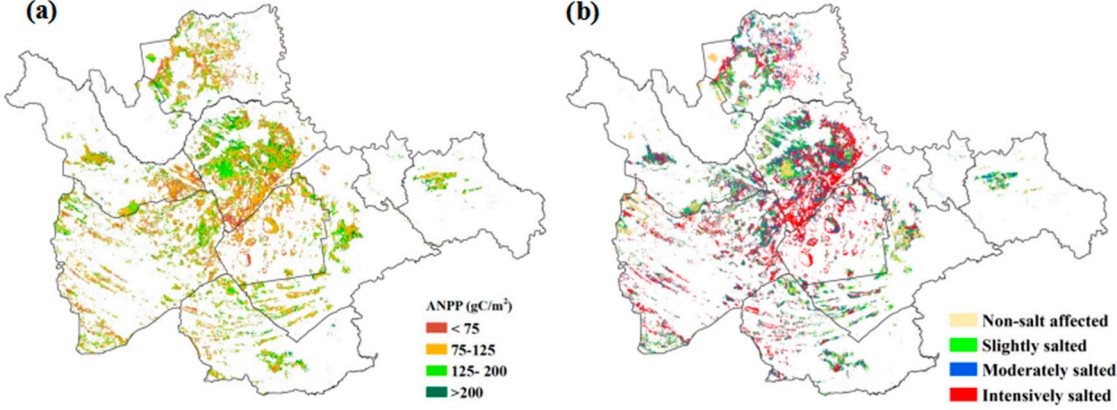

**Figure 7.** Spatial distribution of EC and ANPP in the West Jilin. (**a**) the spatial distribution of ANPP; (**b**) the spatial distribution of soil salinity.

In order to have a clear understanding of the degree of soil salinization in the study area, we ranked four levels for soil salinity using EC. When the value of EC was below 0.2 mS/cm, the region was considered non-salt affected; EC ranging from 0.2 mS/cm to 0.4 mS/cm was considered slightly salted; EC ranging from 0.4 mS/cm to 0.8 mS/cm was considered moderately salted, and EC above 0.8 mS/cm was intensively salted. Figure 7b showed that the non-salt affected area showed scattered distribution, mainly in the central and eastern parts of the region and only accounts for 16.39% of the total grassland

area. Intensively salted regions were concentrated in the low-lying terrain area in the center of the study area, and moderately salted regions were scattered in the middle and south of the area. The salt-affected area was $509.72 \times 10^3$ ha, accounting for 83.61% of the study area; slightly salted was the most common occurrence in salt-affected lands ($116.88 \times 10^3$ ha), accounting for 19.17% of the area. The areas of moderately salted and intensively salted were $183.82 \times 10^3$ ha and $209.02 \times 10^3$ ha, accounting for 30.15% and 34.28% of the grassland, respectively.

### 3.3. Assessment of Grassland Degradation

To effectively estimate grassland degradation, we considered both aboveground vegetation and soil salinity (EC) of grassland. Grassland degradation model was established by combining the change rate of EC and ANPP using the PCA method. The covariance matrix was built from the ground sample data. Through matrix decomposition, we obtained two eigenvalues of 0.738 and 0.083. The first component is defined by the largest eigenvalue 0.738 and the corresponding eigenvector $(-0.261, 0.685)^T$. $F_1$ represents the final data of GDM. The expression of GDM is as follows:

$$GDM = F_1 = -0.224 \times ANPP_{adj} + 0.588 \times EC_{adj} \tag{5}$$

$ANPP_{adj}$ and $EC_{adj}$ are adjusted ANPP and EC, respectively.

The map of grassland degradation was obtained by the above model. The first principal component contains more information than ANPP or EC, respectively. We classified the first component to four levels (non-degraded, slightly degraded, moderately degraded and intensively degraded) referenced on China's grassland degradation classification indicator studies [81] and China's national standard of "parameters for degradation, sandification, and salification of rangelands" (GB19377-2003) [23,28]. The evaluation of non-degraded grassland in the national standard is based on the characteristics of surface vegetation and soil conditions of the same grassland types in the same hydrothermal conditions near the monitoring area [82]. The results showed that the intensively degraded grassland almost coincided with the intensively salted-affected area, and the values of ANPP in the intensively degraded region were less than 75 gC/m$^2$. The slightly degraded areas of grassland were mainly distributed in the moderately salt-affected and slightly salt-affected areas where the values of ANPP above 125 gC/m$^2$. For moderately degraded region of grassland, the values of ANPP ranged from 55 to 125 gC/m$^2$. This part of the grassland was located on the moderately salt-affected and intensively salt-affected areas. The results of statistical analysis show that 61.31% of grassland was degraded; the area was $373.78 \times 10^3$ ha. Figure 8 shows that the non-degraded area of grassland was located in the center and east of the study area, the intensively degraded area was located in the middle and north of the area, the moderately degraded area was scattered throughout the whole region, and slightly degraded area was mainly located on the west of the area. Intensively degraded land was most common in the grassland degraded area, accounting for 34.52%, followed by moderately degraded land. The moderately degraded area was $88.29 \times 10^3$ ha. There was only 12.30% grassland showing as slightly degraded; the area was $75.01 \times 10^3$ ha.

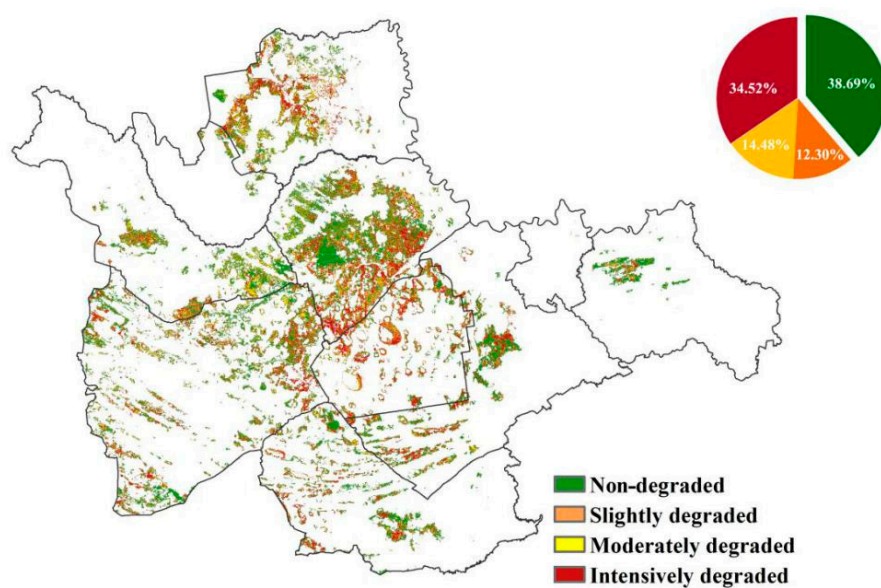

**Figure 8.** Spatial distribution of different grassland degradation degrees over West Jilin in 2017.

## 4. Discussion

### 4.1. Random Forest versus Partial Least Squares Regression

To better demonstrate the advantages of the RF method in predicting grassland degradation parameters, we compared the prediction results of the RF and PLSR methods. The PLSR is an extension of multiple regression analysis, in which the effects of linear combinations of several predictors on a response variable (or multiple response variables) are analyzed. Some researchers used PLSR to estimate the biomass [83] and EC in the past [84,85]. PLSR uses its inferable ability to simulate the possible linear relationship between spectral reflectivity and measured ground data. At the same time, PLSR can also include multiple response variables to effectively deal with strong noise and collinearity of predictive variables [86]. PLSR takes advantage of additional variables or non-linear preprocessing of variables to deal with the non-linearity situation. However, the strategy of using additional factors would cause over-fitting and reduce its real prediction ability [87]. RF could use averaging to improve the predictive accuracy and control over-fitting.

Table 2 shows the result of estimating ANPP and EC using PLSR. For ANPP, the inversion models implied that NDVI and SWIR were the best variables. For the EC model, the most influential variable was the Green band. In our previous study, we found that the Green reflection rose significantly when the EC value was above 1mS/cm, which explained why the inversion results of the power exponent model had high accuracy [39].

Validation data (n = 50) was used to validate the predictive performance of the models. Figure 9 shows the accuracy of the RF and PLSR models. The validation results show that the $R^2$ of the RF models were significantly higher than the PLSR model. Figure 9a,c showed the relationship between actual value and predicted value by RF; Figure 9b,d showed the relationship by PLSR. For ANPP, the $R^2$ increased from 0.57 to 0.87 and from 0.81 to 0.95 for EC. These inversion models which were established by RF had the highest $R^2$ and lowest RMSE. The RF models explained over 85% of ANPP/EC variation, with RMSE of 13.13 gC/m$^2$ and 0.046 mS/cm, respectively. The results indicated that RF is a more suitable technique than PLSR for the estimation and mapping of ANPP and EC.

**Table 2.** The model comparisons of grassland degradation parameters.

| Type of Model | ANPP | $R^2$ | RSME (gC/m$^2$) |
|---|---|---|---|
| Linear | ANPP = 234.561NDVI + 34.067 | 0.540 ** | 20.696 |
| Polynomial | ANPP = 133.798NDVI$^2$ + 153.18NDVI + 43.186 | 0.545 ** | 20.592 |
| Exponential | ANPP = 43.896e$^{2.494NDVI}$ | 0.529 ** | 21.065 |
| LTU | ANPP = 256.824 + 351.608NDVI − 287.695SWIR | **0.571** ** | 20.048 |

| Type of Model | EC | $R^2$ | RSME (mS/cm) |
|---|---|---|---|
| Linear | EC = 4.2862Green − 0.5525 | 0.682 ** | 0.471 |
| Polynomial | EC = 7.1997Green$^2$ − 0.7424Green + 0.0748 | 0.735 ** | 0.437 |
| Power exponent | EC = 3.982Green$^{1.867}$ | **0.738** ** | 0.429 |
| LTU | EC = 6.296Green − 1.982SWIR − 0.186 | 0.703 * | 0.452 |

* represents significance at the 0.05 level; ** represents significance at the 0.01 level; bold number means the maximum value in a column; LTU means linear equation in two unknowns. The unit for ANPP was gC/m$^2$, and unit for EC was mS/cm.

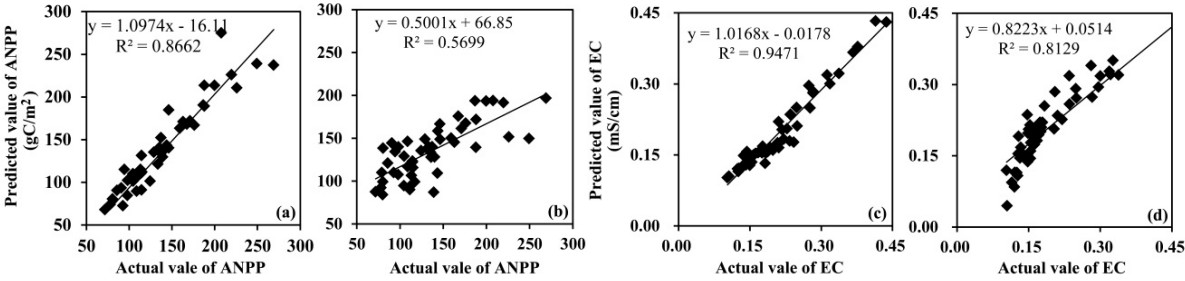

**Figure 9.** Relationships between predicted ANPP/EC and measured ANPP/EC of grassland. (**a**) ANPP model using RF; (**b**) ANPP model using PLSR; (**c**) EC model using RF; (**d**) EC model using PLSR.

RF has integrated learning, both variance and deviation are relatively low, and the generalization performance is superior compared with classical regression trees. RF can provide an effective method to balance the error of datasets when there is an imbalance in classification. Because RF is a tree model, we can use it directly without normalization. Although RF resembles the "black box" method more than PLSR, it still interprets the meaning of decision trees by variable importance [88].

### 4.2. The Importance of Soil Salinity

The west Jilin Province is an ecologically sensitive and fragile zone with a wide area of soil salinization and is one of only three alkali lands in the world. Soil salinization seriously restricts the development of natural steppes and animal husbandry [42]. Soil salinity is a major driving factor of grassland degradation. Numerous grasslands are distributed on the salted soils in west Jilin province, and the issue of grassland soil salinization is attracting increasing attention. With the exacerbation of soil salinization, the exposed proportion of alkali spots significantly increased [39,54]. Soil salinization is one of the major factors restricting the sustainable development of ecology in the west of Jilin province, reducing the production of vegetation, increasing the proportion of non-palatable grassland, and affecting the diversity of vegetation species [22,39]. Figure 10 shows the different statuses of grasslands at different levels of soil salinity; Figure 10a showed the grassland coverage, the coverage rate from high to low is a1 > a3 > a4 > a2, Figure 10b showed the levels of salt-affected soil, the order of soil salinization is b3 > b2 > b4 > b1 from high to low. The soil salt level in Figure 10b3 was higher than other samples; however, the plant coverage was not the lowest. If the grassland degradation model only focused on the grassland cover or yield of grass, this situation would be mistaken for non-degraded areas. In fact, with the increase of soil salinization, the rate of dominant grass species has decreased. GMD overcame the over-reliance on grassland coverage and grass production ability to analyze grassland degradation and produced an effective model which was closer to the actual situation.

Due to soil salinity, 83.61% of grasslands were salt-affected regions. Moreover, the moderately and intensively salted regions were the most common types in salt-affected grassland areas. Therefore, the salinization of grassland is a factor that cannot be ignored in the evaluation of grassland degradation.

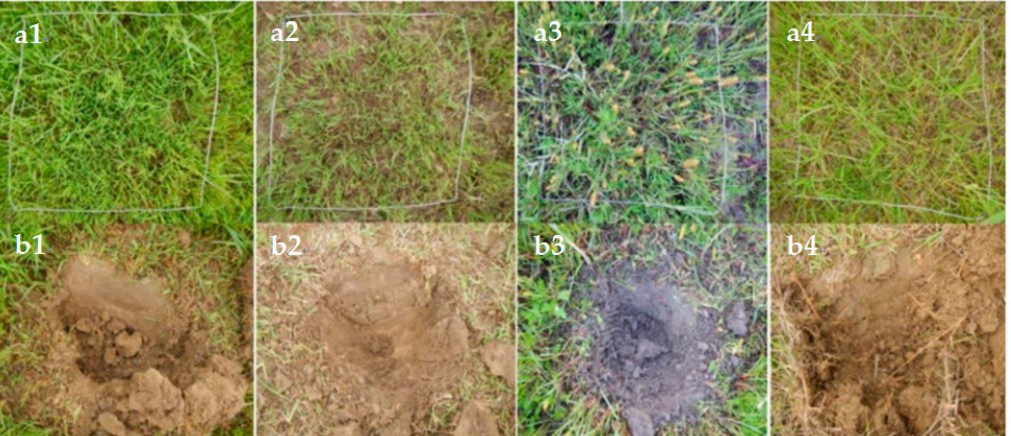

**Figure 10.** The field sampling pictures of the grassland in 2017. (**a1**)–(**a4**) the condition of grass coverage in sampling points; (**b1**)–(**b4**) the soil salinity of the corresponding sampling points.

In past studies, the total area of grassland decreased and grassland was the most severely salt-affected vegetation type in past decades [7,16,39]. Moreover, soil salinity is the key factor of grassland degradation in west Jilin Province [89]. In this paper, we realized that the extremely salted areas had high correlation with strongly degraded grassland. It was found that the health and prosperity of grasslands are highly correlated with soil status; however, this notion is not clearly pointed out in most grassland degradation studies, and soil salinity was seldom integrated into the model calculation of grassland degradation. In previous studies, more attention has been paid to aboveground biomass, plant coverage, and species of vegetation than soil properties in estimating grassland degradation. This groundbreaking study achieves a comprehensive evaluation of grassland degradation from a new point of view.

### 4.3. Uncertainties of Grassland Degradation

Remote sensing offers multiple sensing technologies to monitor grassland degradation. Monitoring grassland degradation via establishing models using measurements and satellite imagery has become much more reliable means compared with traditional methods. There are two major problems facing the assessment of degradation: (i) the uncertainty of baseline assessments and indicator systems and (ii) the misuse of remotely sensed data sources. Liu et al. assessed the grade of grassland degradation using the grass coverage ratio and the proportion of unpalatable grasses, which were sampled by 1 $m^2$ plots; then the risk of degradation model of grassland was obtained by weighted summation with a weight of 0.5 assigned to each parameter [24]. Wei et al. realized the grassland degradation by change rates of grassland cover and NPP, and then quantitatively assessed the contribution of driving forces [8]. Some studies assessed the grassland degradation by interpreting remote imagery with different levels [90]. In these studies, there was some uncertainty in grassland degradation, caused by the selection of degradation indicators and weights allocation of assessment indicators. Moreover, the image resolutions, influence of bands, interpreter's cognition for study area, and professional skills have an impact on the results of grassland degradation. Previous studies show that the spatial distribution of the grassland was primarily affected by the salt in the surface soil over semi-arid or arid regions [37]. In our paper, we selected ANPP and soil salinity as the research parameters of grassland degradation system according to the soil characteristics of west Jilin Province. Moreover, in weight distribution of parameters, we use the PCA method instead of artificial decision to reduce the

uncertainty caused by human interference. Although we have ameliorated the grassland degradation assessment method, there are still some uncertainties. Figure 6 shows the spatial uncertainties of ANPP and EC models and how to reduce the higher RMSE in the south and eastern regions. Further improving the accuracy of grassland degradation is the main focus of our future research.

## 5. Conclusions

In this research, we assessed grassland degradation by not only its coverage but also its quality and productivity. We established a new grassland degradation model, which was specially designed for semiarid regions, based on aboveground net primary production and soil salinity using 166 field samplings. The grassland degradation parameters were assessed by RF verifying the relationship with spectral indices and individual bands from the satellite images. The result shows that the RMSE of ANPP was 15.81 gC/m$^2$ (R$^2$ = 0.85) and that of EC was 0.057 mS/cm (R$^2$ = 0.91). GMD was obtained by estimating ANPP and EC using PAC. GMD revealed 61.31% of grassland was degraded in the study area. Intensively degraded land was most common in the grassland degraded area, accounting for 34.52% of the area, followed by moderately degraded. The moderately degraded area was 88.29 × 10$^3$ ha. Only 12.30% of grassland showed slight degradation, and the area was 75.01 × 10$^3$ ha. Our results demonstrate that our method is a viable solution to grassland degradation monitoring through remote sensing and in-situ data collection of ANPP and EC. In addition, government-aided restoration and protection measures should be implemented in a routine manner, with higher intensity, to minimize future grassland degradation in the west Jilin Province of China.

**Author Contributions:** Hao Yu and Zongming Wang conceived and designed the research; Hao Yu processed the data, performed the experiments and wrote the manuscript draft; Chunying Ren conducted the fieldwork, helped to design the research and reviewed the manuscript; Bai Zhang contributed reagents/ materials/analysis tools; Hao Yu prepared all figures and tables. Zongming Wang, Chunying Ren and Lei Wang helped to conceive the research and reviewed the manuscript.

**Funding:** This research was funded the Strategic Priority Research Program of the Chinese Academy of Sciences (XDA19040500), the Key Project for Field Station Alliance, Chinese Academy of Sciences (KFJ-SW-YW026), the Program Founding from IGA (Y6H2091001), the funding from Youth Innovation Promotion Association Chinese Academy of Sciences (2017277, 2012178), the funding from Jilin Scientific and Technological Development Program (20170301001NY), and funding from the China Scholarship Council (CSC NO. 201804910494).

**Conflicts of Interest:** The authors declare no conflict of interest.

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
