# Peer review of "Using Landsat OLI and Random Forest to Assess Grassland Degradation with Aboveground Net Primary Production and Electrical Conductivity Data"

_ijgi, doi:10.3390/ijgi8110511_

Round 1

Reviewer 1 Report

General comments:

After reviewing the manuscript entitled “Using Landsat OLI and Random Forest to assess 2 Grassland Degradation with Aboveground Net 3 Primary Production and Electrical Conductivity data”, I found the paper readable and informative. I think the readers of IJGI will find the manuscript useful. But, the author (s) need to improve the writing in English. I think the paper should be published with major modifications that makes it acceptable for publication. See the detail comments and English editing in the attached PDF file.

Within the Introduction, to better understand the existence of previous work related to your work, I suggest highlighting with a specific paragraph.

Specific comments and suggestions

See the attached PDF file (ijgi_621312-peer-review_shi.pdf”

Author Response

Dear reviewer,

Thanks for the kind suggestion. We have made sincere efforts in the revision process. We addressed the relationship and difference between previous studies and this study, the advantages and disadvantages of these methods, as well as the reasons for the selection of degradation index, regression method and image selection. We modified the Figures according to the reviewer’s suggestion. The modified parts have been highlighted in the revised manuscript. The manuscript has been polished by two professors and one native speaker, and double checked by the authors. We hope that our revision could satisfy the editorial requirements. Please contact me if you have any concerns or further information is requested. 

Thank you!

The specific response are in the attachment.

Reviewer 2 Report

In this paper the authors implement Random Forest to assess Grassland Degradation using Landsat OLI in the southern Songnen Plains in northeast China. The approach seems interesting to the readers of this journal. I think it can be recommended for publication in IJGI after some changes:

Given the broad and diverse readership of IGJI, I strongly believe readers would better understand what is the object of the paper and why the authors chose one methodology instead of other available options if a more extensive literature review could be offered. As a matter of fact, the whole paper would benefit from a more complete introduction eliciting the following:

a. which other methods are used in literature for similar scopes,
b. advantages and disadvantages of these options
c. for which other applications these methods are used

In conclusion, I am confident that authors performed a valuable work producing and interesting case study.

Author Response

Dear reviewer,

We appreciate the suggestive comments from the reviewer. We have made sincere efforts in the revision process. We strengthened introduction of the methods for grassland degradation assessment at the Introduction section. We also addressed the difference between previous studies and this study, the advantages and disadvantages of these methods, as well as the reasons for the selection of degradation index, regression method and image selection. The modified sentences have been highlighted in the revised manuscript. The revised manuscript has been polished by two professors and one native speaker, and double checked by the authors. We hope that our revision could satisfy the editorial requirements. Please contact me if you have any concerns or further information is requested. Thank you!

The specific responses are in the attachment.
